# Effects of Si Substrates with Variable Initial Orientations on the Growth and Thermoelectric Properties of Bi-Sb-Te Thin Films

**DOI:** 10.3390/nano13020257

**Published:** 2023-01-07

**Authors:** Junze Zhang, Hanwen Xu, Zhuanghao Zheng, Cong Wang, Xinru Li, Fu Li, Ping Fan, Yue-Xing Chen

**Affiliations:** 1Shenzhen Key Laboratory of Advanced Thin Films and Applications, Key Laboratory of Optoelectronic Devices and Systems of Ministry of Education and Guangdong Province, College of Physics and Optoelectronic Engineering, Shenzhen University, Shenzhen 518060, China; 2Hubei Key Laboratory of Low Dimensional Optoelectronic Materials and Devices, Hubei University of Arts and Science, Xiangyang 441053, China

**Keywords:** Bi-Sb-Te, monocrystalline silicon substrate, preferred orientation

## Abstract

For thermoelectric thin film, the substrate plays an important role during the growing process and produces effects on its thermoelectric properties. Some special kinds of substrates provide an optimal combination of influences on both the structure and thermoelectric properties. In this work, Bi-Sb-Te films are deposited on Si substrates with different initial orientations by magnetron sputtering in two ways: with and without a pre-coating process. The preferred orientations of the Bi-Sb-Te films are greatly affected by the substrates, in which the thin film tends to deposit on Si substrate with (100) initial orientation and high (015)-texture, while the (00*l*)-textured Bi-Sb-Te film easily deposits on Si substrate with (110) initial orientation. The experimental and theoretical calculation results indicate that Bi-Sb-Te film with (00*l*)-texture presents good electrical conductivity and a higher power factor than that of film with (015)-texture.

## 1. Introduction

With the consumption of fossil fuel, thermoelectric thin film, which is an emission-free power generation technology, has attracted more and more attention, because of its unique transformation of heat to electric power directly and the flexible characteristic could be applicable to the Internet of things (IoT) and integrated circuit (IC) design [1,2]. The key parameters of thermoelectric thin film are the Seebeck coefficient and electrical conductivity, which determines the power factor (*PF* = *S*^2^*σ*), an evaluation of the output power efficiency of the thermoelectric thin film [3]. With the progressive miniaturization of energy and energy storage devices, this has spurred the development of thin films, which are increasingly replacing bulk materials in applications, as the thin film can better meet the miniaturization and flexibility requirements of its devices [2,4,5,6].

Recently, some new material systems, including Ag_2_Se, Cu_2_Se, Mg_3_Sb_2_ et al. showed outstanding properties and thus performance [7,8,9,10]. For example, Ding et al. reported an n-type Ag_2_Se film on flexible nylon membrane with an ultrahigh power factor ~9.9 μWm^−1^ K^−2^ at 300 K with excellent flexibility [11]. The flexible p-type Cu_2_Se thin TE film with a high (0l0) preferred orientation achieved a maximum *ZT* of 0.42 at 275 °C by the co-sputtering method [8]. However, conventional Sb_2_Te_3_-based materials are still widely studied and extensively used in the commercial field [12,13,14,15]. In virtue of the characteristics of anisotropy for Sb_2_Te_3_ material, the modulation of the preferred orientation growth of the thin films play an important role in optimizing thermoelectric properties. For example, the Sb_2_Te_3_ film with high (00*l*) orientations usually presents better carrier mobility than that of films with (015) preferred orientations, resulting in good electrical transport properties [16,17].

It is widely reported that disparate synthesis approaches and substrates lead to variation in growth preferred orientation of Sb_2_Te_3_-based thin film [18,19,20]. Notably, plenty of substrates could have an influence on the growth of thin film by virtue of differential deposition energy, and lead to the lattice mismatch between the films and substrates [21,22]. Kwon et al. reported on a (001) GaAs wafer used as the substrate for growing Bi_0.4_Sb_1.6_Te_3_ via a metal organic chemical vapor desposition (MOCVD) approach, and a high power factor (*PF*) of 15 μWm^−1^ K^−2^ was achieved [20]. Bi-Sb-Te thin film grown on disarrangement substrates, such as glass or polymerized substrate show a significant difference in performance. For instance, a unique preferred orientation of Bi_0.4_Sb_1.6_Te_3_ deposited on the K9 glass ultimately realized a high power factor over 20.0 μWm^−1^ K^−2^ at room temperature, while Liang et al. reported Bi_0.5_Sb_1.5_Te_3_ film grown on flexible polyimide substrates displayed a relative low value of 2 μWm^−1^ K^−2^ at 475 K [23,24]. Besides, the thermoelectric performance could also be affected by the same substrates with different initial orientations. Guo et al. grew Ni-based superalloy on Ni-based substrates with different initial orientations of (100), (011) and (111) and presented variations in the thermoelectric properties. Additionally, the (001) substrate tends to show stronger epitaxial growth characteristics than others due to the specific lattice arrangement, in which six equivalent crystallographic orientations are arrayed on the substrate surface [25]. For conventional Sb_2_Te_3_ materials, few works have focused on the variations of microstructure and thermoelectric performance by using the same substrate with different initial orientations. Therefore, in this work, we fabricate Bi_0.4_Sb_1.6_Te_3_ films on monocrystalline silicon substrates with three kinds of surface orientations, (100), (110) and (111), respectively. The microstructure and preferred orientation of as-fabricated Bi-Sb-Te films are greatly affected by the initial orientation of Si substrates and the preferred orientation, and the electrical transport properties are investigated systematically. The higher power factor of Bi-Sb-Te films with (00*l*)-texture is further confirmed than that of film with (015) preferred orientation due to its good electrical conductivity [26].

## 2. Materials and Methods

The Bi_0.4_Sb_1.6_Te_3_ target was prepared by combining the mechanical alloying (MA) and spark plasma sintering (SPS) methods; the detail of the synthesis process can be found in our previous work [23]; the composition of the target was detected as Bi_0.41_Sb_1.60_Te_2.99_ by Energy Dispersive X-ray Detector (EDX). Precursor powder was prepared by the ball milling process (QM-3SP2), using high-purity Bi (Aladdin, 99.99%), Sb (Aladdin, 99.99%), and Te (Aladdin, 99.99%) for 10 hours under the protection of an Argon atmosphere. After that, the precursor powder was filled in the graphite die with diameter Φ = 60 mm, then consolidated by SPS (LABOX-325) at 400 °C with a soaking time of 5 min. Three kinds of monocrystalline silicon substrates with orientations of (100), (110), and (111) were used. The monocrystalline silicon substrates with the dimension of 10 × 20 × 1 mm^3^, were cleaned using a semiconductor silicon wafer remote chemical analysis (RCA) cleaning method [27]. The depositing process of Bi_0.4_Sb_1.6_Te_3_ thin film on monocrystalline silicon substrates was employed by the magnetron sputtering technique. The ultimate pressure of the chamber was set as 1.0 × 10^−3^ Pa, and the fixed distance between the target and substrate was 80 mm. Heretofore, we also studied the working vacuum degree, rate of Argon flow, and sputtering power. We controlled the working vacuum at 2 Pa and 40 sccm of the Ar gas flow. The polished Bi_0.4_Sb_1.6_Te_3_ target was set with a DC (direct current) sputtering source, adjusted 15W of power. The thickness of the thin film was controlled by regulating the deposited time. The thickness of Bi-Sb-Te thin film was formed at about 600 nm with the base temperature of 200 °C, and the thin films deposited showed minor component bias from the target. As a routine manipulation of sputtering, the Bi-Sb-Te films were prepared via two different processing approaches, one of which an addition of a pre-coating process is applied. During the pre-coating process, a seed layer (Bi-Sb-Te) of the target film was deposited for 10 min with a thickness of about 100–150 nm. To simplify, two batches of films named as Si(hkl)_S and Si(hkl) were abbreviated for films with the pre-coating process and direct deposition films, separately.

An energy dispersive X-ray spectroscope (EDS) equipped with the scanning electron microscope were used to detected the chemical composition and surface morphology of films (Zeiss, Oberkoche, Germany). The crystal structure and preferred orientations of films were obtained via X-ray diffraction (XRD) equipment (Rigaku, Tokyo, Japan). The Seebeck coefficient and electrical conductivity apparatus (Nezsch, Selb, Germany ) were applied to measure the electrical transport properties of the thin films. The carrier concentration and mobility could be measured by utilizing a Van der Pauw Hall measuring instrument (Nanometrics, Milpitas, CA, USA).

Density functional theory (DFT) calculations were performed to investigate the electrical transport properties of Sb_2_Te_3_ compound along two selected crystallographic planes of (001) and (015). The band structures of Sb_2_Te_3_ [001] and Sb_2_Te_3_ [015] are achieved by using the Vienna Ab Initio Simulation Package (VASP) [28,29,30]. The calculations were done by utilizing the plane-wave pseudopotential formalism. The ion-electron interaction is modeled with the help of the projector-augmented wave (PAW) method for Sb and Te atoms [31]. Cut-off energy for the wave function is set to 450 eV. The generalized gradient approximations (GGA) with the Perdew-Burke-Ernzerhof potential (PBE) are implemented for the exchange-correlation function [32]. The structures are relaxed until forces on each atom become less than 0.10^−5^ eV/Å, and the criterion for energy convergence is set as 1.0 × 10^−6^ eV. The structure of Sb_2_Te_3_ [015] is modeled by 3 × 1 × 1 supercells containing 9 Sb and 6 Se sites. A Monkhorst-Pack k-point mesh of 13 × 13 × 1 is used, including *Γ*-points for the Brillouin zone sampling of the cell in the initial self-consistent field (SCF) step. The electronic energy band is plotted based on a 100 k-points grid.

## 3. Results

Figure 1a shows the XRD patterns of Si(100)_S, Si(110)_S, and Si(111)_S thin film, which engages a pre-coating process to grow a seed layer of Bi-Sb-Te film. All the patterns of Si(100)_S, Si(110)_S, and Si(111)_S films could be well-indexed as the precise phase of Bi_0.4_Sb_1.6_Te_3_ (PDF#72-1836), and no second phase is observed. Due to the different orientations of Si substrates, the preferred orientation of the films is different, which can be clearly seen in the expanded peaks of (003) and (015), as shown in Figure 1b,c. For the Si(100)_S sample, the film exhibits a strong (015) peak, while the other two films show equal intensities between (00*l*) and (015) peaks. This can be attributed to the dissimilar interface energy variation of Bi-Sb-Te films on different substrates, resulting in the diverse lattice mismatches.

To determine the preferential orientations of as-deposited films, the Lotgering factor (*F*), also known as the orientation factor, of (00*l*) and (015) planes is calculated and displayed in Table 1. For example, the computation of the *F*(00*l*) orientation factor can be estimated by the equations:(1)$$F=\frac{P-P_{0}}{1-P_{0}}$$
(2)$$P=\frac{I\left(00l\right)}{\sum I\left(hkl\right)}$$
(3)$$P_{0}=\frac{I_{0}\left(00l\right)}{\sum I_{0}\left(hkl\right)}$$
where $I_{0}\left(00l\right)$ and $\sum I_{0}\left(hkl\right)$ represent the intensity of standard data (PDF#72-1836) of total (00*l*) diffraction intensity and the whole intensity of diffraction, respectively. While $I\left(00l\right)$ and $\sum I\left(hkl\right)$ are the corresponding diffraction intensities of the samples. The range of *F* is between 0 and 1. The closer to 1, the stronger the orientation is. The *F*(015) of Si(100)_S film is 0.74, while values of *F*(015) for Si(110)_S and Si(111)_S are 0.26 and 0.32, respectively, confirming the significant effect of the initial orientations of substrates during growing of the Bi-Sb-Te films.

The electrical transport properties of Si(100)_S, Si(110)_S, and Si(111)_S films are measured at room temperature and shown in Table 1. Due to the different preferred orientation growth of Bi-Sb-Te films, the variation in the thermoelectric properties can be easily distinguished. As shown in Table 1, Si(100)_S films with high (015) orientation present higher *S* and lower *σ* than that of Si(110)_S and Si(111)_S films, which shows the (00*l*) preferred orientations. This result is commonly reported in this system, as the Bi-Sb-Te films with (00*l*) preferred orientation can provide a faster track for transferring carriers, that is higher carrier mobility can be achieved as shown in Table 1. Although the Si(100)_S possesses the better *S*, the unassailable high *σ* of Si(110)_S and Si(111)_S films emerge as the major factor that contributes to higher power factors. However, the *PF* ~1.62 μwcm^−1^ K^−2^ achieved by Si(110)_S film is relatively low in comparison with other reported Bi-Sb-Te films due to its undeveloped electrical transport properties [20,24,33,34].

As mentioned above, the preferred orientations and thermoelectric properties of as-fabricated Bi-Sb-Te films are greatly affected by the initial orientation of Si substrates. However, a seed layer of Bi-Sb-Te film is deposited by the pre-coating progress, which seems to impair the influence of Si substrates during the growth of Bi-Sb-Te films. To further investigate the role of monocrystalline silicon initial orientations, we skip the pro-coating process and deposit the Bi-Sb-Te film at the beginning of sputtering. Figure 2 shows the XRD patterns of the Bi-Sb-Te films, which were directly deposited on the Si (100), (110), and (111) substrates, respectively. As shown in Figure 2, the XRD patterns also correspond with the standard card of Bi-Sb-Te without any second phases. For the preferred orientation of samples, the calculated *F*(015) is 0.54 for Si(100) film, maintaining the high (015) preferred orientation growth on the Si substrate with (100) orientation. The (00*l*) orientation is enhanced, when the Bi-Sb-Te is deposited on Si substrate with (110) orientation, as the F(00*l*) of Si(110) film is 0.67. The enhanced *F*(015) of Si(111) film is also observed in the XRD pattern. These results can be further confirmed in the extended view of (003) and (015) peaks (Figure 2b,c). Hence, the preferred orientations of Bi-Sb-Te films are greatly influenced by both the initial orientations and surface conditions of Si substrates.

Figure 3 shows the FESEM images of surface and cross-section microstructures for Si(100), Si(110), and Si(111) films. For Si(100) film, the grains are stacked in an irregular way, while most of the grains become organized and tiled parallel to the substrate for Si(110) film. These are consistent with our previous results that the morphology of randomly arranged grains is often indicative of (015) growth orientation [35] and the booming of grains parallel to the substrate normally represents an increase in the *F*(00*l*) as described in XRD patterns (Figure 2a). Obeying this morphologically rule, the growth of gains of Si(111) presents fewer grains parallel to the substate agreeing with lower (00*l*) preferred orientation. The cross-section images for three films with similar thickness ~630 nm are shown in Figure 3b,d,f. The grains are closely stacked with variable arrangement, further confirming the different growing condition for Bi-Sb-Te film on these Si substrates.

Based on the above experimental phenomena, we draw the conclusions that the Si(100) substrate tends to grow highly (015)-textured Bi-Sb-Te film while Si(110) substrate is inclined to have (00*l*)-textured ones independent of the pre-coating process. On the other hand, it is widely reported that more deposited energy is needed to grow (00*l*)-textured Bi-Sb-Te film than that of (015)-textured film [17,22]. The Si(100) substrate seems to possess a larger interface lattice mismatch with Bi-Sb-Te structures, leading to the bigger interfacial energy than that of the Si(110) substrate for growing the Bi-Sb-Te film [21]. Hence, to conquer the high interfacial energy, the Si(100) substrate is likely to grow (015)-textured film but not (00*l*)-textured under the same deposition conditions.

In Figure 4, we compare the thermoelectric performance as well as the orientation factors of typical Si(100) and Si(110) films, which present (015)-textured and (00*l*)-textured characteristics, respectively. As shown in Figure 4a, Si(100) film with a dominant (015)-texture possesses a larger *S* (~160 μV/K) and lower *σ* (~70 S/cm) than the Si(100) film, which is in good agreement with the results mentioned above and reported in the previous literature [36]. The remarkable *σ* (~200 S/cm) contributes to a better power factor (~4 μW/cmK^2^). The origin of high *σ* can be easily understood as the carrier mobility of Si(110) film is one order higher than that of the Si(100) sample (Figure 4b), which consists of a highly (00*l*) preferred orientation, as shown in Figure 4c. Furthermore, density functional theory (DFT) calculations were performed to roughly compare the electrical transport behavior of a simple Sb_2_Te_3_ structure, which shares the same crystal structure with the as-fabricated Bi-Sb-Te films, along two selected crystallographic planes of (001) and (015), and the crystal structures can be observed in Figure 4e. The calculated detail can be found in the experimental section and the calculated band structures are presented in Figure 4d,f. By comparing with the band structure of Sb_2_Te_3_ (001), it is observed that the bands near the Fermi level of the Sb_2_Te_3_(015) structure show a flat characteristic, indicating the significant variation in its density of state. This usually leads to an increase in effect mass (*m**) and large Seebeck coefficient [37,38,39]. The calculations indirectly support the fact that Sb_2_Te_3_-based films with preferred orientation of (015) should have a higher Seebeck coefficient than that of (00*l*)-textured film; it is also consistent with the results in Figure 4a. Furthermore, we estimated the *m** of Si(100) and Si(110) films based on the single parabolic band (SPB) model [40]; the *m** values of Si(100) and Si(110) films are 1.31 *m*_0_ and 0.60 *m*_0_, respectively, shown in Figure 4b. The result is also consistent with the experimental *S* in Figure 4a. Based on the SPB model, the thermoelectric materials with high μ*m*_d_*^3/2^ are considered to have high electrical transport properties, where *m*_d_* is estimated by the product of *m** and degree of degeneracy [41]. Hence, the better power factor achieved by Si(110) film is mainly attributed to its high carrier mobility while maintaining decent *S*. This work explores a way to deposit Bi-Sb-Te films with variable preferred orientation on different initial orientational Si substrates by magnetron sputtering. To reveal the correlation between the preferred orientation and electrical transport properties of Sb_2_Te_3_ based films, more exploration is needed in the future.

## 4. Conclusions

To summarize this work, we grew the Bi-Sb-Te films on Si substrates with different initial orientations by using a magnetron sputtering method from the lab-made Bi_0.4_Sb_1.6_Te_3_ target. As a result, with or without a pre-coating process, the highly (015)-textured Bi-Sb-Te film was easier to grow on Si substrate with (100) initial orientation, while the (00*l*)-textured Bi-Sb-Te films tended to grow on Si substrate with (110) initial orientation. Based on the experimental and theoretical calculation results, the film with (00*l*)-texture presented better electrical conductivity than that of (015)-textured films due to its high carrier mobility, resulting in a better power factor. This work provides a way to investigate the effects of the substrate initial orientation on the growth conditions and thermoelectric performance of Sb_2_Te_3_ based thin film.

## Figures and Tables

**Figure 1 nanomaterials-13-00257-f001:**
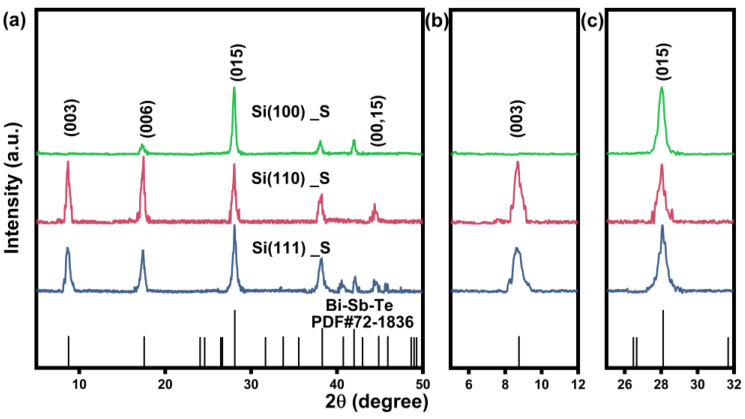
(**a**) XRD patterns of Bi-Sb-Te films deposited on Si substrates with different initial orientations. Magnified XRD patterns of (**b**) (003) and (**c**) (015) peaks.

**Figure 2 nanomaterials-13-00257-f002:**
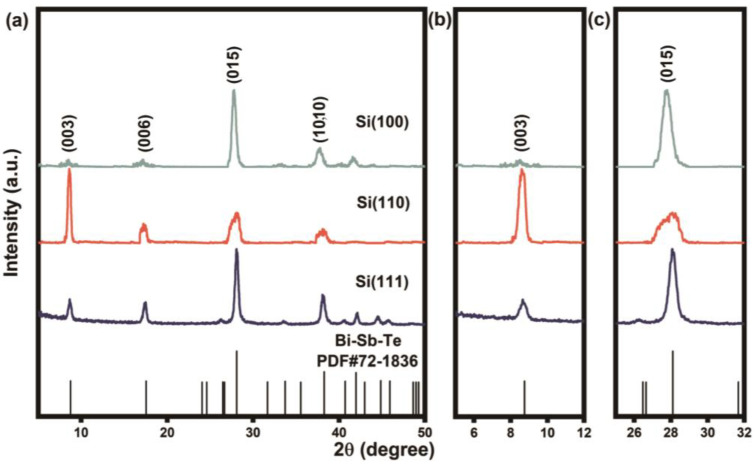
(**a**) XRD patterns of Bi-Sb-Te films deposited on Si(100), Si(110) and Si(111) substrates. (**b**,**c**) Extended view of (003) and (015) peaks.

**Figure 3 nanomaterials-13-00257-f003:**
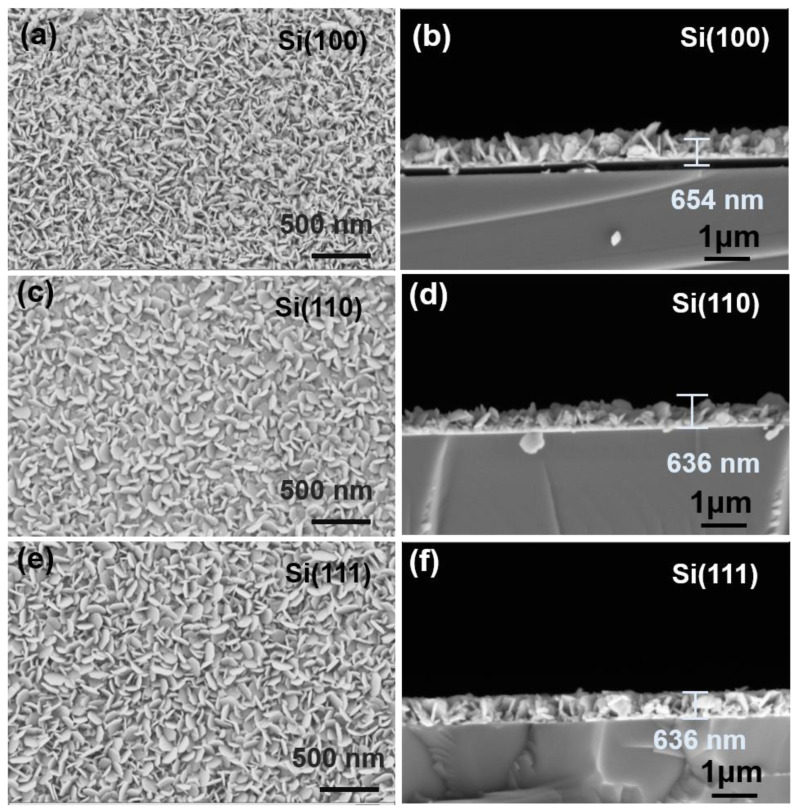
FESEM images of the surface and cross-section microstructures of (**a**,**b**) Si(100), (**c**,**d**) Si(110) and (**e**,**f**) Si(111) Films.

**Figure 4 nanomaterials-13-00257-f004:**
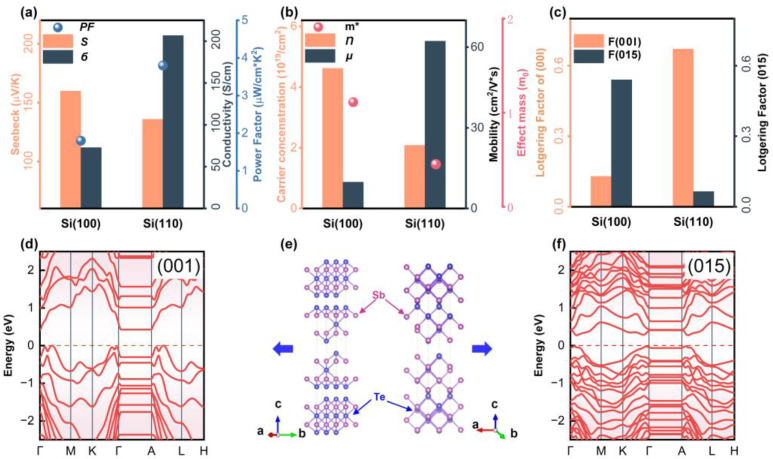
(**a**) Seebeck coefficient *S*, electrical conductivity, *σ* and power factor *PF*; (**b**) Carrier concentration *n*, mobility μ and *m** (**c**) *F*(015), *F*(00*l*) of Si(100), and Si(110) thin films. (**d**,**f**) Electrical band structures of (**e**) two crystal models of Sb_2_Te_3_ based on the electrical transport along planes (001) and (015).

**Table 1 nanomaterials-13-00257-t001:** Room temperature thermoelectric properties and orientation factors of Si(100)_S, Si(110)_S, and Si(111)_S thin films.

Sample	Conductivity (S/cm)	Seebeck (μV/K)	Carrier Concentration (10^19^/cm^3^)	Carrier Mobility (cm^2^/V∗s)	PF (μw/cm∗K^2^)	F (00*l*)	F (015)
Si(100)_S	45	136	3.14	8.96	0.83	0.12	0.74
Si(110)_S	132	111	0.87	94.86	1.62	0.59	0.26
Si(111)_S	116	115	2.27	31.94	1.53	0.48	0.32

## Data Availability

The data presented in this study are available on request from the corresponding author.

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
