# Peer review of "Effects of Si Substrates with Variable Initial Orientations on the Growth and Thermoelectric Properties of Bi-Sb-Te Thin Films"

_nanomaterials, 2023, doi:10.3390/nano13020257_

Round 1

Reviewer 1 Report

Title of the manuscript

Effects of Si substrates with variable initial orientations on the growth and thermoelectric properties for Bi-Sb-Te thin films

Manuscript Number: nanomaterials-2124283

Comments

In this work, thin films of Bi-Sb-Te are deposited on Si substrate with different orientations with magnetron sputtering. The film with (00l) orientation results in good thermoelectric performance over (015) textured. The work is comprehensive and well-executed. I offer the following comments and would recommend this work for publication in nanomaterials after satisfactory revision.

1. It is important to consider changes in the thermal conductivity with the different orientations and therefore, I strongly recommend including thermal conductivity results and corresponding zT values and related discussion.

2. As compared to the literature where this work stands? I suggest comparing current results with the literature and presenting them in a tabulated form.

Recommendation Minor revision

Thank you

Author Response

Dear Editor and Reviewers,

  Thank you very much for your reviews on our submitted manuscript entitled with “Effects of Si substrates with variable initial orientations on the growth and thermoelectric properties for Bi-Sb-Te thin films”. We appreciate the referees’ comments and suggestions, and have tried our best to revise the manuscript closely to what was advised. Please find below our detailed replies to referees’ comments as well as our revised manuscript where the modified parts are highlighted.

Reviewer #1

In this work, thin films of Bi-Sb-Te are deposited on Si substrate with different orientations with magnetron sputtering. The film with (00l) orientation results in good thermoelectric performance over (015) textured. The work is comprehensive and well-executed. I offer the following comments and would recommend this work for publication in nanomaterials after satisfactory revision.

Response:We gratitude the positive suggestions for improvement of this work. The comments were extremely helpful for us to adapt to journal’s requirements.

  1. It is important to consider changes in the thermal conductivity with the different orientations and therefore, I strongly recommend including thermal conductivity results and corresponding zT values and related discussion.

Response:Thank you for the suggestions. The thermal conductivity of the thin film is hard to measure, and the trend measurement relies on deposition on the chip and using a ‘3ω’ method which could replace the substrate as a key role in this research.     Therefore, we appreciate that the reviewer can understand the lack of the thermal conductivity values and estimated ZTs.

In this work, we mainly focus on the effects of Si substrates with different initial orientations on the growth of thin film, and the preferred orientations of the Bi-Sb-Te films are greatly affected by the substrates. The experimental and theoretical calculation results indicates that as-fabricated Bi-Sb-Te film with (00l)-textured presents good electrical conductivity and higher power factor than that of film with (015)-textured.

  1. As compared to the literature where this work stands? I suggest comparing current results with the literature and presenting them in a tabulated form.

Response: Thanks for your suggestions. Based on the main purpose of this work that   the relationship between the thermoelectric properties and the substrate with initial orientations are investigated. Actually, there are few similar works focus on this issue, in the following figure, we compare the (00l) orientations factors with other reported work and found that the Bi-Sb-Te film in our work show highly (00l) orientations. Furthermore, we also grown highly (015) orientation film (F=0.67) as shown in Figure 4c, then we can investigate the electrical transport properties of films with different dominate orientations.

Figure R1 Comparison of (00l) orientation factors with reported literatures of Bi0.4Sb1.6Te3 samples.

Reviewer 2 Report

This manuscript studies the effects of Si substrate orientation on the preferred growth direction of Bi-Sb-Te films with or without a pre-coating process. The authors also used DFT simulation to understand the mechanism behind better electrical conductivity for the Bi-Sb-Te film with (00l)-textured over the film with (015)-textured. The manuscript is well drafted. I would only recommend the authors to consider the following 2 comments:

1. Please provide more details about seed layer. In the “Experiment”, it only mentioned that “During the pre-coating progress, a seed layer of target film were deposited for 10 mins.” What is the material for the seed layer? What method is used to deposit such layer? How thick is such seed layer? Any effect of seed layer’s thickness of final coating’s texture orientation?

2. The starting target is Bi0.4Sb1.6Te3. However, there is no composition identified for the deposited thin films, which are generalized called by the authors as “Bi-Sb-Te”. Can the authors determine the actual stoichiometry of the thin films with a combination of EDS, XPS and XRD techniques?

3. The DFT simulation in Fig. 4(d, e, f) is based on Sb2Te3, while the experimental results in Fig. 1, 2, 3 and 4 (a-c) are based on Bi-Sb-Te” thin films. Although I agree that both Sb2Te3 and “Bi-Sb-Te” thin films share the same lattice structures, their lattice parameters and atom arrangement are different, which affects the band structures for each material. Hence, I suggest that the authors (1) determine the composition of “Bi-Sb-Te” thin films and (2) use its own lattice structure parameters for DFT simulation.

BTW, the authors should provide the detailed structure (i.e., rhombohedral) with lattice parameters used for DFT for the readers.

Author Response

Reviewer #2

This manuscript studies the effects of Si substrate orientation on the preferred growth direction of Bi-Sb-Te films with or without a pre-coating process. The authors also used DFT simulation to understand the mechanism behind better electrical conductivity for the Bi-Sb-Te film with (00l)-textured over the film with (015)-textured. The manuscript is well drafted. I would only recommend the authors to consider the following 2 comments:

Response:We appreciate the positive comments to this work. The comments were extremely helpful to further improve our manuscript.  

  1. Please provide more details about seed layer. In the “Experiment”, it only mentioned that “During the pre-coating progress, a seed layer of target film were deposited for 10 mins.” What is the material for the seed layer? What method is used to deposit such layer? How thick is such seed layer? Any effect of seed layer’s thickness of final coating’s texture orientation?

Response:Thank you for your suggestion. The seed layer comes as a routine operation of magnetron sputtering, which could adjust the power output in process. In this work, according to our experimental design, the seed layer deposition aimed to get a better lattice match with substrate. The seed layer was also Bi-Sb-Te and deposited via magnetron sputtering, and the thickness is about 100-150nm. Because the seed layer couldn’t achieve a better properties, the thickness of seed layer didn’t got us attention. We had added more experimental detail including the thickness of seed layer in the revised manuscript and highlighted. 

  1. The starting target is Bi0.4Sb1.6Te3. However, there is no composition identified for the deposited thin films, which are generalized called by the authors as “Bi-Sb-Te”. Can the authors determine the actual stoichiometry of the thin films with a combination of EDS, XPS and XRD techniques?

Response: We appreciate reviewer to point out the potential risk of misleading readers by using ‘Bi-Sb-Te’ as the thin film. Generally, the magnetron sputtering of the compound target could cause a minor component bias, so we didn’t use Bi0.4Sb1.6Te3 under a careful consideration. We also add a EDS data to confirm the stoichiometry of the thin film as deposited (Bi:8.22 at%, Sb:32.25 at%, Te:59.53 at%). The related text had been added in manuscript and highlighted.

  1. The DFT simulation in Fig. 4(d, e, f) is based on Sb2Te3, while the experimental results in Fig. 1, 2, 3 and 4 (a-c) are based on “Bi-Sb-Te” thin films. Although I agree that both Sb2Te3 and “Bi-Sb-Te” thin films share the same lattice structures, their lattice parameters and atom arrangement are different, which affects the band structures for each material. Hence, I suggest that the authors (1) determine the composition of “Bi-Sb-Te” thin films and (2) use its own lattice structure parameters for DFT simulation. BTW, the authors should provide the detailed structure (i.e., rhombohedral) with lattice parameters used for DFT for the readers.

Response: We thank for the information given by the reviewer. In this work, we used the DFT simulation from two different orientations to counterevidence the preferred orientation could actually affect the band structure roughly, finally impact on the Seebeck coefficient and the electrical conductivity. The calculation results agree with the experiment data. In DFT simulation, the structure of Sb2Te3(015) is modeled by 3*1*1 supercells containing 9 Sb and 6 Se sites, which could simply meet our demands, while the Sb2Te3(015) model is cut from the original Sb2Te3 structure few atoms. Since the Sb2Te3 shares the same crystal structure with Bi0.4Sb1.6Te3, the simulation performed on the (00l) and (015) planes is considered to have similar  electric transport properties behaviors between Sb2Te3 and Bi-Sb-Te structures.  

Reviewer 3 Report

The authors have described the effect of thermoelectric properties and growth of Bi-Sb-Te film on several oriented substrates. This manuscript is interesting to the potential readers but there are some questions that need to be addressed prior to the publication as this manuscript show lack of characterization and discussion of the films. The authors have introduced SEM and XRD results as characterization techniques but it is hard to see the difference between these samples.

-       -  Please introduce the abbreviation and symbol before their first use. For example what is MOCVD, PF and Φ?

-          Can the authors explain why (00l) orientations presents better carrier mobility?

-          Is the dimension of substrate really 10*20*1 mm and what is RCA cleaning?

-          It is not clear from the Figure 1 about the orientation as all of these films show (00L) orientation? Only some films show (015) orientation. Anyway what is this (00,15) peak?

-          Prior to XRD measurements, are these films aligned because it seems it is coupled XRD measurement.

-          If the authors mentioned in the introduction that (00l) orientation is ideal situation but here in the table 1 (00L) is worse?

-          The discussion of SEM analysis is weak as the authors can not compared SEM results with XRD results. TEM analysis is more preferable as it can show the orientation of the films.

-          The discussion part is still missing?

Author Response

Reviewer #3

The authors have described the effect of thermoelectric properties and growth of Bi-Sb-Te film on several oriented substrates. This manuscript is interesting to the potential readers but there are some questions that need to be addressed prior to the publication as this manuscript show lack of characterization and discussion of the films. The authors have introduced SEM and XRD results as characterization techniques but it is hard to see the difference between these samples.

Response: Thanks for the positive reviews, and valuable suggestions. In our previous work[23], we have already obtained an ideal environment to growth the Bi-Sb-Te thin film. XRD patterns already indexed well as the standard pattern(PDF#72-1836), so the samples show difference only on the intensity of all peaks. The SEM cross-section images also shows the layer-shaped grains with different angle between the substrate which could corresponding the Lotgering factor (F) of samples.

  1. Please introduce the abbreviation and symbol before their first use. For example what is MOCVD, PF and Φ?

Response: We are sorry for the ill-consideration of using abbreviation in the manuscript. In the revised manuscript, we added the full meaning of these symbols for first use in text.

MOCVD (Metal Organic Chemical Vapor Desposition); PF (Power Factor); Φ (Diameter of the graphite die).

  1. Can the authors explain why (00l) orientations presents better carrier mobility?

Response: This unique phenomenon primary caused by the special crystal structure of Bi-Sb-Te. Carrier could get a better mobility via in-plane orientation, without the barrier of the grain boundary, this results can be also found in more reported works[16, 17].

  1. Is the dimension of substrate really 10*20*1 mm and what is RCA cleaning?

Response: Yes, the monocrystalline silicon substrate was incised as the shape mentioned. The RCA(Remote Chemical Analysis) cleaning is a standard cleaning process of the silicon wafer, and reviewer could obtain the details in the reference [27].

  1. It is not clear from the Figure 1 about the orientation as all of these films show (00L) orientation? Only some films show (015) orientation. Anyway what is this (00,15) peak?

Response: Not all Si(hkl)_S samples show a strong intensity of (00l) orientation in figure one, and we also didn’t make a statement about all these films show (00l) orientation either. For Si(100)_S sample, the film exhibits strong (015) peak, while the others show equally intensities between (00l) and (015) peaks. Besides, we calculated the Lotgering factor of each samples’ XRD intensities,as shown in table1, which could give a proof support our expression. The (00,15) peak is also a characteristic peak of Bi0.4Sb1.6Te3, which is included in the range of (00l) plane representation.

  1.  Prior to XRD measurements, are these films aligned because it seems it is coupled XRD measurement.

Response: The XRD measurements obtained via X-ray diffraction (XRD, Ultima IV, Rigaku), used a 10 degree per minute scanning rate and the range was 8 to 50 degree. We guaranteed that all the data is untreated, and under consideration of data presentation in a figure, the intensities just normalized.

  1.  If the authors mentioned in the introduction that (00l) orientation is ideal situation but here in the table 1 (00L) is worse?

Response: I am afraid the reviewer misunderstanding the information in table 1. The samples Si(110)_S and Si(111)_S shown larger orientation factor of (00l) than Si(100)_S, and former with (00l) preferred orientation can provide a faster track for transferring carriers which could be proved by the properties of carrier mobility and conductivity.

7. The discussion of SEM analysis is weak as the authors can not compared SEM results with XRD results. TEM analysis is more preferable as it can show the orientation of the films.The discussion part is still missing?

Response: Thanks for the suggestion of the advanced measurement. We thought that the TEM measurement is unnecessary, because the XRD shown the difference of the samples and the SEM cross-section image could also seen the layer-by-layer crystal grain with different angle between the substrate. Empirically speaking, the TEM analyze exists more unpredictable situation, and we had already obtained the same preferred orientation phenomena through crystal-scale (XRD measurement) and macro-scale (SEM measurement). So the TEM test is no need as far as we concerned.

Round 2

Reviewer 1 Report

Thank you for taking the time to revise the manuscript by considering both comments. Although thermal conductivity measurement and estimation of the zT are pivotal, their lack does not affect the discussion and conclusion of the study.  Therefore, I would like to recommend this article for publication.

Reviewer 3 Report

The authors have improved the manuscript based on the reviewers questions. It is now suitable for the publication.